# Characterizing Employees with Primary and Secondary Caregiving Responsibilities: Informal Care Provision in Malaysia

**DOI:** 10.3390/healthcare11142033

**Published:** 2023-07-16

**Authors:** Caryn Mei Hsien Chan, Ching Sin Siau, Jyh Eiin Wong, Noorazrul Yahya, Nor Aniza Azmi, Shin Ying Chu, Mahadir Ahmad, Agnes Shu Sze Chong, Lei Hum Wee, Jo Pei Tan

**Affiliations:** 1Centre for Community Health Studies, Faculty of Health Sciences, Universiti Kebangsaan Malaysia, Kuala Lumpur 50300, Malaysia; caryn@ukm.edu.my (C.M.H.C.); chingsin.siau@ukm.edu.my (C.S.S.); wjeiin@ukm.edu.my (J.E.W.); mahadir@ukm.edu.my (M.A.); agnes@ukm.edu.my (A.S.S.C.); 2Center for Diagnostic, Therapeutic and Investigative Studies, Faculty of Health Sciences, Universiti Kebangsaan Malaysia, Kuala Lumpur 50300, Malaysia; azrulyahya@ukm.edu.my (N.Y.); noraniza.azmi@ukm.edu.my (N.A.A.); 3Centre for Healthy Ageing and Wellness, Faculty of Health Sciences, Universiti Kebangsaan Malaysia, Kuala Lumpur 50300, Malaysia; chushinying@ukm.edu.my; 4School of Medicine, Faculty of Health and Medical Sciences, Taylor’s University, Lakeside Campus, No. 1, Jalan Taylor’s, Subang Jaya 47500, Malaysia; 5Medical Advancement for Better Quality of Life Impact Lab, Taylor’s University, 1 Jalan Taylors, Subang Jaya 47500, Malaysia; 6Department of Social Care and Social Work, Faculty of Health and Education, Manchester Metropolitan University, Manchester M15 6BH, UK; j.tan@mmu.ac.uk

**Keywords:** informal caregiving, middle generation, sandwich generation, employees, caregiving, childcare, eldercare, Asian, Malaysia, organizational support

## Abstract

There is a need to determine the extent to which Malaysian employees reconcile both paid employment and informal care provision. We examined data from the Malaysia’s Healthiest Workplace via AIA Vitality Online Survey 2019 (N = 17,286). A multivariate multinomial regression was conducted to examine characteristics for the following groups: primary caregiver of a child or disabled child, primary caregiver of a disabled adult or elderly individual, primary caregiver for both children and elderly, as well as secondary caregivers. Respondent mean age ± SD was 34.76 ± 9.31, with 49.6% (*n* = 8573), identifying as either a primary or secondary caregiver to at least one child under 18 years, an elderly individual, or both. Males (*n* = 6957; 40.2%) had higher odds of being primary caregivers to children (OR 2.06; 95% CI 1.85–2.30), elderly (OR 1.24; 95% CI 1.09–1.41) and both children and elderly (OR 1.87; 95% CI 1.57–2.22). However, males were less likely to be secondary caregivers than females (OR 0.61; 95% CI 0.53–0.71). Our results highlight the differences in characteristics of employees engaged in informal care provision, and to a lesser degree, the extent to which mid-life individual employees are sandwiched into caring for children and/or the elderly.

## 1. Introduction

Aging populations and long-term care are issues faced by both developed and developing societies [1]. The aging megatrend in Southeast Asia, one of the most rapidly aging regions globally, has led to challenges in care provision and support. With more than 20% of people over age 65 by 2040, Southeast Asian societies like Malaysia will experience soaring health and care demands [2]. According to the Malaysian National Health and Morbidity Survey (NHMS), 5.7% of adults in the general population reported the provision of unpaid care over the past year [3]. The extent to which individuals holding paid employment are also burdened with providing informal care remains unclear [4]. The load of care on employees who are also caregivers is concerning since they must balance work demands and various caregiving responsibilities [5].

Of primary concern is the burden of care on individuals simultaneously balancing work demands and multiple caregiving responsibilities, which entails having both childcare and eldercare responsibilities. In some cases, this care extends to providing for parents or parents-in-law and children. This is mainly seen among mid-life individuals who may be sandwiched into the need to care for multiple generation households, which is endemic to Asian families [6]. This is a common enough arrangement that it is known as the sandwich or middle generation, which refers to a generation of individuals who are responsible for the upbringing of their children as well as for the care of their aging parents or parents-in-law. 

The sandwich generation thereby plays a dual caregiving role and makes a substantial contribution to care support and provision, which can be in the form of financial support or care coordination. However, a shrinking working-age population and increased longevity are deemed unsustainable for long-term care provision and their burden is well-documented [7]. In addition, longer life expectancies, increasing disease burdens and extended retirement ages are likely to escalate this issue in the near future. Mid-life individuals may likely have to work for far longer years to support themselves and their aging parents, and possibly their adult children as well [8]. This issue has not borne closer scrutiny, despite its possible associations with contemporary organizational culture and wider societal norms in the Malaysian context.

In Malaysia, home-based informal care is far more common than institutionalized care, with family members bound by the sacred ties of familial duty and obligation due to locoregional emphasis on collectivistic cultural values and norms [9]. In collectivist cultures, especially within Southeast Asia, families are typically characterized by deferment to parental authority and filial piety and responsibility [10]. There is a need to examine how mid-life individuals, who are in paid employment, provide informal care in this part of the world. Thus, the present study aimed to examine characteristics of employees with childcare and or eldercare responsibilities (the sandwich generation), differentiated by primary, secondary or no care provision.

## 2. Methods

The data for this study was sourced from the AIA 2019 Malaysia’s Healthiest Workplace online survey conducted in 2019. This cross-sectional survey applied a two-stage stratified non-random sampling among 230 organizations partaking in Malaysia’s Healthiest Workplace survey. The stratification was performed by company size (large/medium/small). All employees from these organizations were invited to participate in the survey. In this study, we examined adults aged 18 and above with complete key demographic and socioeconomic details. The analysis included complete responses to questions related to their health and informal caregiving role.

Data were collected from May to July 2019. Participation in the survey entailed responding to a questionnaire online. The tenets of the 2013 Declaration of Helsinki were followed throughout the conduct of the study. Consent from the respondents was obtained before they started the survey. The study was conducted in accordance with the Declaration of Helsinki, and the survey protocol was approved by the Research Ethics Committee of the Universiti Kebangsaan Malaysia (JEP-2019-692). 

### 2.1. Caregiving Status

Participants were asked to indicate whether or not they had caregiving responsibilities. An answer of ‘yes’ indicated an individual having one or more caregiving responsibilities as follows: (1) primary carer of a child, (2) primary carer of a disabled child, (3) primary carer of a disabled adult, (4) primary carer of an elderly adult of ≥65 years and above, and/or (5) secondary carer (non-specific). Respondents who were not involved in direct or indirect care provision, including monetary support, were categorized as having no caregiving responsibilities (6). 

Caregiving status was reclassified and differentiated. Original categories (1) and (2) were merged to create the composite category of primary caregiver of a child or disabled child, while (3) and (4) were merged to form the category of primary caregiver of a disabled adult or older individual. An additional category where individuals identified as primary caregivers of both child(ren) and elderly were identified. The categories of secondary caregiver (non-specific) and no caregiving responsibilities were maintained. 

Aside from the standard demographic characteristics of age, gender, marital status and educational attainment, this study also took into account occupation type, income, psychological distress, fatigue and comorbid medical condition. 

Occupation type followed the World Health Organisation classification [11]. Income approximated division according to low, middle and high-income demarcation lines [12]. Psychological distress was assessed using the Kessler psychological distress scale (K10) [13], with the applied cut-off score of ≥20 considered to indicate high psychological distress. The presence of fatigue was assessed via the single-item question, “During your waking time, do you feel tired or fatigued?”. Comorbid medical condition was indicated by the presence of at least one of the following conditions: bronchial asthma, heart conditions, diabetes mellitus, hypertension, migraines, arthritis, musculoskeletal disorders and mental illness.

### 2.2. Statistical Analysis

Bivariate analysis was used to describe the profile of the sandwich generation in Malaysia, segregated by care provision status. A multinomial logistic regression was then performed to examine characteristics associated with the primary caregivers of either child(ren) or the elderly, primary caregiver of both child(ren) and elderly and secondary caregivers (unspecified), with those with no caregiving responsibilities as the comparison group. All analyses were controlled for demographic and socioeconomic covariates. A *p* < 0.05 was considered significant. All analyses were conducted using SPSS version 28. 

## 3. Results

Table 1 shows the characteristics of the overall sample of employees. Close to half the sample (*n* = 8573; 49.6%) reported assuming caregiving responsibilities as either primary or secondary caregivers to children and/or adults. Respondent age ranged from 18 to 88, with a mean age ± SD of 34.76 ± 9.31. Close to two-thirds of the sample were female (*n* = 10,329; 59.8%). Of the overall sample, 41% were single. Most reported the possession of an undergraduate degree (*n* = 9016; 52.2%) and were employed in largely executive, administrator or senior manager positions (*n* = 4448; 25.7%). By income, the highest numbers (*n* = 6636; 38.4%) clustered around RM 3999 or less income bracket (equivalent to approximately USD 950). The majority of respondents reported no significant psychological distress (*n* = 10,661; 61.7%) but indicated significant fatigue (*n* = 11,205; 64.8%) and the presence of at least one comorbidity (*n* = 14,831; 85.8%).

Table 2 indicates the results of a multivariate multinomial logistic regression for primary caregivers of a child, primary caregivers of an elderly individual and primary caregivers for both children and the elderly as well as secondary carers. Primary caregivers of children had higher odds of being in the age brackets of 18–24 years (OR 2.07; 95% CI 1.28–3.35) and 25–34 years (OR 4.44; 95% CI 3.27–6.03). Primary caregivers of older adults had lower odds of being in the age brackets of 18–24 years (OR 0.09; 95% CI 0.06–0.14) and 25–34 years (OR 0.40; 95% CI 0.29–3.65) but higher odds of being in the age bracket of 45–54 years (OR 2.34; 95% CI 1.73–3.18). Secondary caregivers also had lower odds of being in the age brackets of 18–24 years (OR 0.37; 95% CI 0.24–0.57) and 25–34 years (OR 0.59; 95% CI 0.41–0.84) but higher odds of being in the age bracket of 45–54 years (OR 2.04; 95% CI 1.43–2.89).

Sex, specifically being male, was significantly associated with being the primary caregivers of children (OR 2.06; 95% CI 1.85–2.30), elderly individuals (OR 1.24; 95% CI 1.09–1.41) and for both children and elderly persons (OR 1.87; 95% CI 1.57–2.22). Males, however, had lower odds of being secondary caregivers compared to females (OR 0.61; 95% CI 0.53–0.71). Being single was less associated with caregiving responsibilities for either children (OR 0.01; 95% CI 0.01–0.01) or both children and the elderly (OR 0.05; 95% CI 0.03–0.08). Being married predicted secondary caregiving (OR 2.53; 95% CI 1.61–3.97). 

Primary caregivers for children had higher odds of reporting no formal education or lower than secondary school completion (OR 1.25; 95% CI 1.01–1.55) or post-secondary school completion (O-level equivalent) (OR 1.23; 95% CI 1.01–1.47). Primary caregivers of the elderly had slightly higher odds of post-secondary school completion (O-level equivalent) (OR 1.42; 95% CI 1.14–1.76). Primary caregivers of both children and the elderly also had slightly higher odds of post-secondary school completion (O-level equivalent) (OR 1.33; 95% CI 1.01–1.76). Secondary caregivers were more likely to have no formal education or lower than secondary school completion (OR 0.70; 95% CI 0.52–0.94).

By occupation type, primary caregivers of adults or older individuals had higher odds of employment in clerical and administrative support occupations (OR 1.58; 95% CI 1.14–2.17), as well as sales type occupations (OR 1.62; 95% CI 1.12–2.35). By income, employees who had household incomes of RM3,999 or less had lower odds of providing care to the elderly (OR 0.73; 95% CI 0.59–0.92) or caregiving for both children and elderly (OR 0.50; 95% CI 0.37–0.68).

Caregivers of the elderly had higher odds of reporting psychological distress (OR 1.18; 95% CI 1.04–1.35). All categories of primary caregivers as well as secondary caregivers reported a higher likelihood of comorbidities compared to individuals who professed no caregiving responsibilities.

## 4. Discussion

Caregiving has traditionally been associated with females [14]. At a glance, therefore, our finding that males were more likely than females to assume responsibility for primary caregiving is surprisingly obverse. An underlying gender perspective should, however, be taken into account here. A possible explanation could be that males equate providing financial upkeep with caregiving. The literature on informal caregiving has shown a potential explanation is that males are often reported as the main caregivers, but the actual daily work and instrumental support are provided by females (i.e., wife/daughter in law/daughter). Married males may be receiving help from their partner/wife when taking on more responsibilities as primary caregivers. On the other hand, married females may only be able to be the secondary caregiver as they are expected to care for their parents-in-law [15,16]. This is often revealed in qualitative research which further explore the experience of caregiving [15,17]. Females who were primary or secondary caregivers might be less likely to be employed and therefore their responses may not have been captured in this survey [18,19].

In this sample, there were more female than male respondents. These high female employment rates are consistent with Malaysia’s female labour force participation rate, which stood at 55.6% in 2019 [20]. Another plausible explanation is related to the industry that took part in this survey. For example, hospitality and service sectors tend to have a higher percentage of female than male employees. One could argue the possibility that this means that females could have been more obliging toward requests to complete this survey. But one could also argue that females in active employment were less likely to be married or have a child and had absolved themselves of, or at least partially relinquished, familial responsibilities in order to be as competitive as their male counterparts. This highlights the issue of female job participation. It may be those females, on whom the onus of day-to-day primary caregiving falls, are less likely to hold formal employment. This, in part, reflects the lack of options around early childhood care with only sporadic formal public services available, especially for infants and toddlers aged 0–3. 

Married people are more likely to have children and have in-laws to care for. In Malaysia, living arrangements that include multiple generations of an extended family are the norm, although the extended family household is decreasing (27.8% in 1980 to 18% in 2010) and the single household is increasing (55.2% in 1980 to 70.2% in 2010) [17]. Members of the extended family tend to include parents-in-law. Thus, having multiple generations of family living in a single household is common in Malaysia as it reflects how Malaysians traditionally and culturally live as a function of filial piety. Stagnant wages, rising living expenses and lowered purchasing power mean that this arrangement may not be by choice, but because it is the only way an individual can afford to live. With the large social benefits and efficacies of cost/work and chores/time, multigenerational households make absolute economic sense. In Western societies, such living arrangements occur in difficult times, as they essentially ensure free babysitting and homeschooling as well as lower mortgages and insurance since housing is shared. While we still have a considerable number of multigeneration household in Malaysian society, it could be argued that the three-generation household is no longer sustainable for urban living due to rising housing costs and limited space. Commuting parents is another common phenomenon among mid-life individuals with multiple care responsibilities and frequent commuting between two cities as part of informal solution for rising child-care issues.

Employees who reported caring for a child or disabled child were more likely to be individuals from the age brackets of 18–24 and 25–34 years. Primary caregivers to children were also more likely to possess no to minimal formal education. Thus, the care for minors seems to fall squarely upon individuals from the 18–24 and 25–34 age bracket with little to no education, who continue to hold full-time employment while assuming caregiving duties. We should, therefore, look into continuing education programs (i.e., affordable, part-time, distance-learning) to help these individuals increase their educational attainment and thereby secure wider and better paying job options.

Individuals in the 18–24 and 25–34 age brackets were 91% less likely to report being the primary caregiver of a disabled adult or elderly individual. Instead, this responsibility was up to 2.34 times more likely to be assumed by those from the 55 and above age bracket. A plausible reason for this could be that individuals from the younger age group may not be able to assume the care responsibilities for elderly as they lack the time and financial resources to support them. In this study, lower-income individuals were less likely to assume caregiving responsibilities, likely due to their financial capacities. 

Taking care of either child(ren) or the elderly can be tedious. Caring for both is likely more so. With improved life expectancies, we are likely to see an increase in the elderly population, with adult children needing to assume or take on the burden of responsibility for caring for their elderly parents (the assumption here is that older age is associated with reduced ability to live independently) in addition to their growing children while simultaneously needing to work. The concerning question here is that as individuals in the 35–44 and 45–54 age bracket grow older, who will care for them, wedged as they are between filial and parental duties? Put into context, as of 2019, the oldest millennial was 38 at the time of the survey. In the US, 66% of older millennials have children. We cannot possibly ascertain the number of millennials in Malaysia who are parents, but as the youngest millennials age and join their ranks, it is likely that many more of them will go on to assume the role of dual care provider/caregiver.

Caregiving for a disabled adult or elderly individual was associated with psychological distress among employees in this sample. Established factors that contributed to psychological distress among caregivers, such as caring for persons needing a higher level of care (e.g., patients with cancer or individuals with a neurodevelopmental disorder), were not measured in this study. In fact, being a secondary caregiver appeared to be protective of psychological distress. This finding appears to support the idea that giving care can be rewarding, with its own mental health benefits [21]. One possible explanation is that secondary caregivers are likely a step removed from the kind of burden faced by primary caregivers to children and adults, as the former do not have to take full responsibility.

Primary caregivers of adults or older individuals were more likely to be employed in clerical and administrative support occupations, as well as sales-type occupations. The former is likely attributable to the structure and regular working hours associated with caregiving as it allows planning. It may be easier to schedule medical follow-ups for elderly dependents when one has a job in clerical and administrative support. Caregivers of elderly or disabled adults may also be more likely to pick sales occupations due to the flexible nature of the job compared to other occupation types.

Fatigue was prevalent across the board, with close to two-thirds reporting significant fatigue, although this appeared unassociated with caregiving in this sample. However, comorbid illness was significantly associated with both primary and secondary caregivers, who were 1.2–1.6 times more likely than individuals who reported no caregiving responsibilities to report comorbidities. Other studies have shown that informal caregiving contributes to poorer physical health [18,22,23]. This may likely be due to the amount of attention, time and effort put into providing care for others, resulting in the neglect of their own wellbeing. 

Compared to only 5.7% of adults in the general population who reported provision of unpaid care [3], a far higher number in this sample provided care to children or elderly dependents. This may be due to a difference in the way caregiving is defined in this study. In the Malaysian National Health and Morbidity Survey (2020), caregiving referred to providing care for individuals with long term health conditions, such as stroke, kidney disease and dementia [24], whereas caregiving in our study refers to caring for children of the elderly, regardless of their health status. An inherent limitation of our study was that financial support was not differentiated from the provision of care.

The prevalence of informal caregiving in our study is comparable with another study in the US where 9% of females aged 45–56 reported simultaneously caring for their parent and dependent children [25]. This prevalence rose substantially to over 30% when adult children aged 18 and over were factored in as dependents by individuals of both genders aged 35–75, who reported the provision of time or money to parents and children [26]. These rates are consistent with research on the Boomerang Generation, which indicated that almost a third of young adults aged between 25–34 live with their parents [27]. In Ireland, 31% of females aged 50–69 reported being sandwiched between their parents and children, with two-thirds of this sample simultaneously providing grandchild care [28]. It is important to consider that informal caregiving rates might vary considerably across nations with similar sociodemographic profiles.

## 5. Conclusions

The recession, pandemic and shifting demographics are intensifying the pressures on the “sandwich generation”—those supporting both children and parents. Moving beyond the prevalence of multiple role occupancy, we need to explore how juggling multiple roles influences caregiving activities, labour force participation and health/wellbeing. Our findings posit several implications for consideration in terms of policy. The findings from this research are of direct relevance for policymakers and practitioners in Malaysia concerned with the provision of long-term care to its ageing population due to shifting demographics.

## Figures and Tables

**Table 1 healthcare-11-02033-t001:** Characteristics of Malaysian employees by caregiving status (N = 17,286).

Characteristics		Caregiving Status					N (%)	*p*-Value
	Primary Caregiver of Both Children and Adults *n* = 881 (%)	Primary Caregiver to a Child *n* = 4814 (%)	Primary Caregiver to an Adult or Older Person *n* = 1570 (%)	Secondary Caregiver (Nonspecific) *n* = 1308 (%)	No Caregiving Responsibilities *n* = 7310 (%)	Prefer Not to Say *n* = 1403 (%)		
Age categories								0.001
18–24 years	8 (0.9)	43 (0.9)	52 (3.3)	84 (6.4)	1630 (22.3)	149 (10.6)	1966 (11.4)	
25–34 years	149 (16.9)	1649 (34.3)	596 (38.0)	527 (40.3)	4162 (56.9)	605 (43.1)	7688 (44.5)	
35–44 years	452 (51.3)	2150 (44.7)	475 (30.3)	413 (31.6)	945 (12.9)	344 (24.5)	4779 (27.6)	
45–54 years	241 (27.4)	883 (18.3)	347 (22.1)	217 (16.6)	360 (4.9)	234 (16.7)	2282 (13.2)	
55 years and above	31 (3.5)	89 (1.8)	100 (6.4)	67 (5.1)	213 (2.9)	71 (5.1)	571 (3.3)	
Sex								0.001
Female	446 (50.6)	2398 (49.8)	973 (62.0)	991 (75.8)	4716 (64.5)	805 (57.4)	10,329 (59.8)	
Male	435 (49.4)	2416 (50.2)	597 (38.0)	317 (24.2)	2594 (35.5)	598 (42.6)	6957 (40.2)	
Marital status								0.001
Single	35 (4.0)	53 (1.1)	878 (55.9)	455 (34.8)	5202 (71.2)	472 (33.6)	7095 (41.0)	
Married	795 (90.2)	4535 (94.2)	590 (37.6)	811 (62.0)	1869 (25.6)	744 (53.0)	9344 (54.1)	
Other—cohabitating, separated, divorced, widowed	43 (4.9)	202 (4.2)	56 (3.6)	29 (2.2)	118 (1.6)	35 (2.5)	483 (2.8)	
Prefer not to say	8 (0.9)	24 (0.5)	46 (2.9)	13 (1.0)	121 (1.7)	152 (10.8)	364 (2.1)	
Education								0.001
No formal education or lower than 0-levels completion	105 (11.9)	708 (14.7)	228 (14.5)	131 (10.0)	694 (9.5)	356 (25.4)	2222 (12.9)	
A-levels or equivalent	218 (24.7)	1065 (22.1)	372 (23.7)	284 (21.7)	1223 (16.7)	354 (25.2)	3516 (20.3)	
Undergraduate degree	387 (43.9)	2277 (47.3)	708 (45.1)	699 (53.4)	4402 (60.2)	543 (38.7)	9016 (52.2)	
Postgraduate degree	171 (19.4)	764 (15.9)	262 (16.7)	194 (14.8)	991 (13.61)	150 (10.7)	2532 (14.6)	
Occupational categories								0.001
Executive, administrator, or senior manager	376 (42.7)	1556 (32.3)	527 (33.6)	356 (27.2)	1386 (19.0)	247 (17.6)	4448 (25.7)	
Professional	214 (24.3)	1230 (25.6)	336 (21.4)	343 (26.2)	2305 (31.5)	272 (19.4)	4700 (27.2)	
Clerical and administrative support	22 (2.5)	132 (2.7)	71 (4.5)	26 (2.0)	344 (4.7)	80 (5.7)	675 (3.9)	
Sales occupation	10 (1.1)	135 (2.8)	52 (3.3)	40 (3.1)	207 (2.8)	66 (4.7)	510 (3.0)	
Technical support/technician or junior professional	62 (7.0)	475 (9.9)	125 (8.0)	120 (9.2)	796 (10.9)	124 (8.8)	1702 (9.8)	
Service occupation	100 (11.4)	676 (14.0)	245 (15.6)	258 (19.7)	1041 (14.2)	273 (19.5)	2593 (15.0)	
Elementary and Combined all other	97 (11.0)	610 (12.7)	214 (13.6)	165 (12.6)	1231 (16.8)	341 (24.3)	2658 (15.4)	
Income								0.001
≤RM 3999	127 (14.4)	1209 (25.1)	452 (28.8)	469 (35.9)	3776 (51.7)	603 (43.0)	6636 (38.4)	
RM 4000 to RM 7999	290 (32.9)	1762 (36.6)	558 (35.5)	453 (34.6)	2108 (28.8)	334 (23.8)	5505 (31.8)	
≥RM 8000	361 (41.0)	1416 (29.4)	423 (26.9)	300 (22.9)	910 (12.4)	180 (12.8)	3590 (20.8)	
Prefer not to disclose	103 (11.7)	427 (8.9)	137 (8.7)	86 (6.6)	516 (7.1)	286 (20.4)	1555 (9.0)	
Psychological distress								
Yes	281 (31.9)	1458 (30.3)	635 (40.4)	418 (32.0)	3275 (44.8)	558 (39.8)	6625 (38.3)	
No	600 (68.1)	3356 (69.7)	935 (59.6)	890 (68.0)	4035 (55.2)	845 (60.2)	10,661 (61.7)	
Fatigue								0.001
Yes	515 (58.5)	2921 (60.7)	957 (61.0)	856 (65.4)	5066 (69.3)	890 (63.4)	11,205 (64.8)	
No	366 (41.5)	1893 (39.3)	613 (39.0)	452 (34.6)	2244 (30.7)	513 (36.6)	6081 (35.2)	
Comorbid illness								0.001
Yes	773 (87.7)	4178 (86.8)	1387 (88.3)	1177 (90.0)	6147 (84.1)	1169 (83.3)	14,831 (85.8)	
No	108 (12.3)	636 (13.2)	183 (11.7)	131 (10.0)	1163 (15.9)	234 (16.7)	2455 (14.2)	

Note: *p*-values are chi-squared values indicating differences between caregiver groups by characteristic. Percentages should be interpreted by column.

**Table 2 healthcare-11-02033-t002:** Multinomial regression of employee characteristics associated with caregiving status.

	Primary Caregiver of a Child or Disabled Child *n* = 4367			Primary Carer of a Disabled Adult or Older Person *n* = 1401			Primary Caregiver of Both Child(ren) and Older Person *n* = 772			Secondary Caregiver (Nonspecific) *n* = 1212		
	Sig.	Exp(B)	95% CI	Sig.	Exp(B)	95% CI	Sig.	Exp(B)	95% CI	Sig.	Exp(B)	95% CI
Age												
18–24 years	0.003	2.072	1.281–3.349	<0.001	0.091	0.059–0.140	0.953	1.027	0.428–2.461	<0.001	0.367	0.237–0.567
25–34 years	<0.001	4.441	3.273–6.025	<0.001	0.399	0.290–0.549	0.290	1.300	0.799–2.116	0.003	0.587	0.412–0.838
35–44 years	<0.001	13.127	9.755–17.664	0.087	1.298	0.963–1.749	<0.001	8.168	5.225–12.769	0.002	1.720	1.228–2.408
45–54 years	<0.001	9.209	6.770–12.528	<0.001	2.341	1.725–3.177	<0.001	6.808	4.319–10.731	<0.001	2.032	1.429–2.889
55 years and above (Ref.)	-	-	-				-	-	-	-	-	-
Sex												
Male	<0.001	2.064	1.853–2.299	0.001	1.240	1.090–1.411	<0.001	1.865	1.568–2.218	<0.001	0.612	0.526–0.712
Female (Ref.)	-	-	-				-	-	-	-	-	-
Marital status												
Single, never married	<0.001	0.009	0.006–0.014	0.503	1.129	0.791–1.612	<0.001	0.047	0.028–0.080	0.235	0.756	0.477–1.199
Married	0.005	1.438	1.113–1.858	0.659	0.924	0.650–1.314	0.096	1.382	0.944–2.025	<0.001	2.530	1.613–3.967
Other—cohabitating, separated, divorced, widowed (Ref.)	-	-	-	-	-	-	-	-	-	-	-	-
Educational attainment												
No formal education or lower than secondary school completion	0.041	1.251	1.009–1.550	0.113	1.231	0.952–1.591	0.933	1.015	0.717–1.436	0.019	0.703	0.523–0.944
Post-secondary school completion	0.027	1.228	1.023–1.473	0.001	1.417	1.144–1.757	0.042	1.333	1.011–1.757	0.654	1.056	0.833–1.337
Undergraduate degree	0.166	1.112	0.957–1.292	0.339	0.916	0.766–1.096	0.561	0.934	0.740–1.177	0.312	1.105	0.911–1.340
Postgraduate degree (Ref.)	-	-	-	-	-	-	-	-	-	-	-	-
Occupation type												
Executive, administrator, or senior manager	0.815	0.978	0.815–1.174	0.378	1.102	0.888–1.367	0.434	1.129	0.833–1.530	0.331	1.126	0.886–1.429
Professional	0.344	0.918	0.769–1.096	0.051	0.807	0.651–1.001	0.558	0.911	0.665–1.246	0.377	1.107	0.884–1.386
Technician or junior professional	0.168	1.165	0.938–1.448	0.968	1.005	0.774–1.306	0.813	1.047	0.714–1.537	0.057	1.304	0.992–1.715
Clerical and administrative support	0.705	1.061	0.781–1.441	0.005	1.577	1.144–2.175	0.427	1.249	0.721–2.163	0.274	0.775	0.492–1.223
Service occupation	0.981	0.998	0.827–1.204	0.229	1.148	0.917–1.439	0.984	0.997	0.711–1.397	0.003	1.429	1.130–1.808
Sales occupation	0.099	1.312	0.951–1.812	0.010	1.624	1.124–2.346	0.473	0.770	0.377–1.572	0.003	1.855	1.240–2.774
Combined all other (Ref.)	-	-	-	-	-	-	-	-	-	-	-	-
Income												
Less than RM 1000 to RM 3999	0.055	0.836	0.696–1.004	0.007	0.733	0.585–0.919	<0.001	0.499	0.366–0.680	0.087	1.236	0.970–1.575
RM 4000 to RM 7999	0.328	1.077	0.928–1.249	0.891	1.013	0.845–1.214	0.145	0.849	0.682–1.058	0.032	1.245	1.019–1.521
RM 8000 and above (Ref.)	-	-	-	-	-	-	-	-	-	-	-	-
Psychological distress												
Yes	0.097	0.909	0.812–1.018	0.013	1.184	1.037–1.353	0.116	1.160	0.964–1.396	<0.001	0.727	0.629–0.839
No (Ref.)	-	-	-	-	-	-	-	-	-	-	-	-
Fatigue												
Yes	0.288	1.063	0.950–1.189	0.068	0.882	0.770–1.010	0.898	1.012	0.845–1.211	0.072	1.143	0.988–1.322
No (Ref.)	-	-	-	-	-	-	-	-	-	-	-	-
Comorbid medical condition												
Yes	0.006	1.234	1.062–1.434	<0.001	1.412	1.168–1.706	0.014	1.380	1.068–1.784	<0.001	1.574	1.272–1.947
No (Ref.)	-	-	-	-	-	-	-	-	-	-	-	-

Note: *p* is considered significant at <0.05. All analyses were controlled for demographic and socio-economic covariates. Reference group: no caregiving responsibilities (N = 6715).

## Data Availability

The data underlying this article will be shared on reasonable request to AIA Malaysia.

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
