# Peer review of "Characterizing Employees with Primary and Secondary Caregiving Responsibilities: Informal Care Provision in Malaysia"

_healthcare, 2023, doi:10.3390/healthcare11142033_

Round 1

Reviewer 1 Report

Dear authors, thank you for providing this window into the everyday of Malaysian workers, caregivers or not. Based in another country it is interesting for me to compare your results with other studies. There are some problems you need to address: 

The simplest is the English and some typos, for example l 240 extra "individual" and l 197 "upon" and unclear things or expressions, such as l 204 "single household". Upon reading I realise this can be three (or more) generations, sounds odd in Western ears. Maybe you can find other words for the different household types? (Yes, probably tricky). Are there any one-person households in Malaysia?

But the main problem is that you use the word care: this usually means more or less hands-on help-support-care (yes!). It seems you also include financial support in your concept care?! I then find your 49 % caregivers to be low... I guess most males see themselves as (the) breadwinner of the family/household? If money and similar kinds of support is involved in your definition of care, I suggest you change the title and reasoning in the text. Maybe something like "Caregiving and support by employees in Malaysia" IF you can not separate financial support and "real" caregiving..  If the survey has separate information on care and financial (indirect) support respectively, it would be interesting to see an analysis of only caregiving...  I notice the 5,7 % in another survey who stated that they were caregivers. I have seen rates for the adult population in different surveys at around 12-16 % (Spain) and 18-24 % (Sweden) and various rates in between or even higher, with less strict definitions. This you need to discuss in the text. 

You need a native English-speaker to check your language

Author Response

Point 1: The simplest is the English and some typos, for example l 240 extra "individual" and l 197 "upon" and unclear things or expressions, such as l 204 "single household". Upon reading I realise this can be three (or more) generations, sounds odd in Western ears. Maybe you can find other words for the different household types? (Yes, probably tricky). Are there any one-person households in Malaysia?

Response 1: We are grateful that you have pointed out the errors. The extra ‘individual’ in line 240 and extra ‘upon’ has been corrected and removed. We have also substituted the unnecessarily specific ‘three generations’ with ‘multiple generations’ to reflect the norm in this setting, although certainly there are one-person households in Malaysia.

Point 2: But the main problem is that you use the word care: this usually means more or less hands-on help-support-care (yes!). It seems you also include financial support in your concept care?! I then find your 49 % caregivers to be low... I guess most males see themselves as (the) breadwinner of the family/household? If money and similar kinds of support is involved in your definition of care, I suggest you change the title and reasoning in the text. Maybe something like "Caregiving and support by employees in Malaysia" IF you can not separate financial support and "real" caregiving.  If the survey has separate information on care and financial (indirect) support respectively, it would be interesting to see an analysis of only caregiving...  I notice the 5,7 % in another survey who stated that they were caregivers. I have seen rates for the adult population in different surveys at around 12-16 % (Spain) and 18-24 % (Sweden) and various rates in between or even higher, with less strict definitions. This you need to discuss in the text. 

Response 2: We are unfortunately unable to differentiate financial support from provision of actual care due to the way the question was worded. This is now indicated as a limitation of this study.

Comments on the Quality of English Language

Point 3: You need a native English-speaker to check your language

Response 3: The manuscript has been proofread to ensure grammatical accuracy.

Reviewer 2 Report

Thank you for letting my read this interesting article: Characterizing employees with primary and secondary caregiving responsibilities: informal care provision in Malaysia. For me it was interesting to read your article. As you wrote in the "discussion", we are used to the female, not the male in the careing situation.

The tables could have been simplified, however I guess the "lay out" will be improoved.

Comments: The abstract gives the reader an important background for an important article

A multivariate multinomial regression was conducted to examine characteristics for the following groups: primary caregiver of a child or disabled child, primary caregiver of a disabled adult or elderly individual, primary caregiver for both children and elderly, as well as secondary caregivers.

However, males were less likely to be secondary caregivers than females (OR 0.61; 95% CI 0.53-0.71). Our results highlight the differences in characteristics of employees engaged in informal care provision, and to a lesser degree, the extent to which mid-life individual employees are sandwiched into caring for children and/or the elderly.

 1. Introduction

 L. 53 The authors use the concept “sandwich” generation. This makes a

substantial contribution to care provision.

Comment: this concept could be exemplified a bit more.

 Suggestion for minor “corrections”

L. 57 - 58…. with mid-life individuals likely having to work for far

longer years to support themselves and their aging parents and

possibly their adult children as well. 8 ………………..

L. 59 intersectionality between contemporary organizational culture and wider societal norms in the Malaysian context.

L. 66 There is a need to examine how mid-life individuals in paid

employment provide informal care in this part of the world.

2. Methods

L. 105 scale(K10) change to scale (K10)

L. 127 By income, the highest numbers (n = 6,636; 38.4%) clustered

around the less than RM3, 999 income brackets (~USD = 950).

Please, it could be adding a few explaining words, what is RM3?

Page 8. nr. (L. 227-28) Primary caregivers of a disabled adult or

elderly individual were up to 91% less likely to be assumed by those

aged 18-24 and 25-34, but up to 2.34 times more likely to be as – 228

summed by those from the 55 and above age bracket.

L. 250. In fact, being a secondary caregiver appeared to be protective

of psychological distress. Explain

3. Results

Table 1-2 shows the characteristics of the overall sample of employees.

4. Discussion

The data from the studies was interesting. In this article the male,

not the females were responsibility for primary care. The author

concluded that in their study.

References: The authors have included 30 articles in their references from the year of 2000 – 2022.

Conclusion: The data is informative; however, it is heavy to read. The

discussion chapter gave the reader a better understanding of the study

that their data were relevance for policymakers and practitioner in Malaysia

Author Response

Point 1: The tables could have been simplified, however I guess the "lay out" will be improoved.

Response 1: We acknowledge that the tables look somewhat cramped due to the need to confirm to the template. However, once the manuscript has been typeset during production, a better layout will likely ensue for the tables.

Comments: The abstract gives the reader an important background for an important article

A multivariate multinomial regression was conducted to examine characteristics for the following groups: primary caregiver of a child or disabled child, primary caregiver of a disabled adult or elderly individual, primary caregiver for both children and elderly, as well as secondary caregivers.

However, males were less likely to be secondary caregivers than females (OR 0.61; 95% CI 0.53-0.71). Our results highlight the differences in characteristics of employees engaged in informal care provision, and to a lesser degree, the extent to which mid-life individual employees are sandwiched into caring for children and/or the elderly.

Point 2:

  1. Introduction
  2. 53 The authors use the concept “sandwich” generation. This makes a substantial contribution to care provision.

Comment: this concept could be exemplified a bit more.

Response 2: This concept has been briefly elaborated.

Point 3: Suggestion for minor “corrections”

  1. 57 - 58…. with mid-life individuals likely having to work for far longer years to support themselves and their aging parents and possibly their adult children as well. 8 ………………..
  2. 59 intersectionality between contemporary organizational culture and wider societal norms in the Malaysian context.
  3. 66 There is a need to examine how mid-life individuals in paid employment provide informal care in this part of the world.

Response 3: We have edited L57-59. For L66, we have added to explanation: ‘Mid-life individuals may likely have to work for far longer years to support themselves and their aging parents, and possibly their adult children as well’ for better comprehension purposes’.

Point 4: 2. Methods

  1. 105 scale(K10) change to scale (K10)
  2. 127 By income, the highest numbers (n = 6,636; 38.4%) clustered

around the less than RM3, 999 income brackets (~USD = 950).

Please, it could be adding a few explaining words, what is RM3?

Page 8. nr. (L. 227-28) Primary caregivers of a disabled adult or

elderly individual were up to 91% less likely to be assumed by those

aged 18-24 and 25-34, but up to 2.34 times more likely to be as – 228

summed by those from the 55 and above age bracket.

  1. 250. In fact, being a secondary caregiver appeared to be protective

of psychological distress. Explain

Response 4: For line 105, the spacing has been added before the parathesis. For line 127, RM3,999 refers to the poverty line threshold, which is equivalent to approximately USD 950. For lines 227-228, the sentence has been revised to make it clearer. For line 250, the explanation for this finding are in the following sentences in that giving care can be rewarding, with its own mental health benefit. Another possible explanation is that secondary caregivers are likely a step removed from the kind of burden faced by primary caregivers to children and adults as the former do not have to bear full responsibility. Secondary caregivers, compared to primary caregivers, do not assume main responsibility in terms of finances or the bulk of the burden of care.

  1. Results

Table 1-2 shows the characteristics of the overall sample of employees.

Point 5: 4. Discussion

The data from the studies was interesting. In this article the male,

not the females were responsibility for primary care. The author

concluded that in their study.

Response 5: Thank you for sharing that you found this interesting.

References: The authors have included 30 articles in their references from the year of 2000 – 2022.

Point 6: Conclusion: The data is informative; however, it is heavy to read. The

discussion chapter gave the reader a better understanding of the study

that their data were relevance for policymakers and practitioner in Malaysia

Response 6: Thank you for your comments and time in providing us valuable feedback in improving the quality of this paper.

Round 2

Reviewer 1 Report

Thank you, it is now passable, as you have now pointed out more clearly that "care" also may mean just economic support. Normally, care means helping someone with things they can not do themselves, but care is admittedly a somewhat vague concept... 

Good Luck with your work!